Multimodal Alzheimer’s disease classification through ensemble deep random vector functional link neural network

Henríquez Pablo A. 1 pablo.henriquez@udp.cl
Araya Nicolás 2 3
1 Departamento de Administración, Universidad Diego Portales , Santiago , Chile
2 Escuela de Informática y Telecomunicaciones, Universidad Diego Portales , Santiago , Chile
3 Department of Computer Science, Pontificia Universidad Católica de Chile , Santiago , Chile
Saibene Aurora
Electronic publication date: 2024 Dec 13
Publication date: 2024
Volume: 10
Electronic Location ID: e2590
Received 2024 Aug 22; Accepted 2024 Nov 18
Copyright: © 2024 Henríquez and Araya
Copyright year: 2024
Copyright holder: Henríquez and Araya
License: This is an open access article distributed under the terms of the Creative Commons Attribution License, which permits unrestricted use, distribution, reproduction and adaptation in any medium and for any purpose provided that it is properly attributed. For attribution, the original author(s), title, publication source (PeerJ Computer Science) and either DOI or URL of the article must be cited.
License URL: https://creativecommons.org/licenses/by/4.0/

Keywords: Alzheimer’s disease, Multimodal machine learning, Random Vector functional link neural networks

Funding: Chilean National Agency for Research and Development (ANID) ANID/PIA/ANILLOS ACT210096 ANID FONDECYT Iniciación 11230396 This work was supported by the Chilean National Agency for Research and Development (ANID) through ANID/PIA/ANILLOS ACT210096; ANID FONDECYT Iniciación Grant 11230396. The funders had no role in study design, data collection and analysis, decision to publish, or preparation of the manuscript.

==============================
Alzheimer’s disease (AD) is a condition with a complex pathogenesis, sometimes hereditary, characterized by the loss of neurons and synapses, along with the presence of senile plaques and neurofibrillary tangles. Early detection, particularly among individuals at high risk, is critical for effective treatment or prevention, yet remains challenging due to data variability and incompleteness. Most current research relies on single data modalities, potentially limiting comprehensive staging of AD. This study addresses this gap by integrating multimodal data—including clinical and genetic information—using deep learning (DL) models, with a specific focus on random vector functional link (RVFL) networks, to enhance early detection of AD and mild cognitive impairment (MCI). Our findings demonstrate that ensemble deep RVFL (edRVFL) models, when combined with effective data imputation techniques such as Winsorized-mean (Wmean), achieve superior performance in detecting early stages of AD. Notably, the edRVFL model achieved an accuracy of 98.8%, precision of 98.3%, recall of 98.4%, and F1-score of 98.2%, outperforming traditional machine learning models like support vector machines, random forests, and decision trees. This underscores the importance of integrating advanced imputation strategies and deep learning techniques in AD diagnosis.

Introduction

Alzheimer’s disease (AD) stands out as the leading cause of dementia in the elderly population in various countries. For example, according to Alzheimer’s Association (2021), it is the 5th leading cause of death in the United States. This disease, with a complex pathogenesis and occasionally of hereditary origin, is characterized by the loss of neurons and synapses, as well as the presence of senile plaques and neurofibrillary degeneration from an anatomical perspective. From a clinical standpoint, it manifests as dementia with a gradual onset and slow progression. Typically, symptoms begin with failures in recent memory and advance until the patient is bedridden, experiencing complete dependence (Breijyeh & Karaman, 2020). In its early stage, the changes experienced by an affected individual may go unnoticed. However, over time, symptoms such as memory loss and language problems begin to emerge, as a consequence of the aforementioned. Furthermore, as the disease progresses, it increasingly affects daily tasks such as speaking or chewing. To date, there is no cure for Alzheimer’s disease. However, diagnosing Alzheimer’s disease early allows for better control of the patient and provides necessary treatments to cope with the disease (Ganaie & Tanveer, 2022). Over the years, various studies have been conducted to gain a deeper understanding of the disease for the subsequent development of treatments and preventive measures in patients (James & Bennett, 2019; Munoz & Feldman, 2000). Furthermore, multiple measurement approaches have been undertaken for the early detection of Alzheimer’s disease. Examples include the analysis of the receiver operating characteristic (ROC) curve (Wu et al., 2013), the whole-brain hierarchical network (Liu et al., 2018a), and the combination of measures such as cortical grey matter volume, cortical thickness, and subcortical volume (Liu et al., 2018b). However, there is currently significant effort being invested in acquiring in-depth knowledge and understanding of the symptoms and progression of Alzheimer’s disease. For this reason, several organizations have undertaken the task of summarizing clinical information from patients to explore various aspects of the disease, such as progression, genomic information, and symptoms, among others (LaMontagne et al., 2019; Marcus et al., 2007, 2010; Koenig et al., 2020; Petersen et al., 2010; Malone et al., 2013; Birkenbihl et al., 2021; Ellis et al., 2009; Beekly et al., 2004). The tools employed in these studies, the use of machine learning algorithms has played a significant role in the quest to characterize all aspects of the disease. Examples of this include the analysis of clinical data, brain imaging, and biological markers using machine learning algorithms (Irfan, Shahrestani & Elkhodr, 2023; Venugopalan et al., 2021; Al Olaimat et al., 2023). These studies aim not only to comprehend the symptoms and progression of Alzheimer’s disease but also to identify patterns, risk factors, and potential biomarkers that may contribute to early and accurate diagnosis. Clinically, in the context of Alzheimer’s disease, patients are classically grouped into one of the following three stages: cognitively normal (CN), mild cognitive impairment (MCI), and AD dementia. MCI is considered a high-risk, transitional stage between healthy aging and dementia. Future clinical decline toward dementia is considered to be more predictable in those with MCI than in CN individuals (Rosenberg et al., 2013; Feldman et al., 2004). However, predicting between CN and MCI patients remains a challenge due to the slight variations that exist between the two types of patients, making it a challenging classification task.

With the advancement in machine learning models, various uses of these models have been proposed for the diagnosis of AD (Moradi et al., 2015; Wang et al., 2021). On the other hand, random vector functional link neural networks (RVFL) have also been utilised for AD (Dai et al., 2017). Other examples include support vector machines (SVM), artificial neural networks (ANN), and deep learning (DL). These are popular machine learning tools used for the diagnosis of AD (Tanveer et al., 2020; Li & Zhao, 2023, Fathi, Ahmadi & Dehnad, 2022). Models based on SVM and ANN exhibit significant differences in their optimization problems: the former provides the global optimal solution, and the latter converges to the local optimal solution (Cortes & Vapnik, 1995). In both SVM and ANN models, the extraction of significant features is an important step. The efficiency of SVM and ANN depends on the efficiency of the functions. Among neural networks, deep learning or representation learning has been the recent trend. Deep learning automates the process of feature extraction through a hierarchy of different feature representations (Ganaie & Tanveer, 2022).

Deep learning architectures consist of multiple layers of neurons arranged in a hierarchical manner, facilitating the extraction of both simple and complex features. The innermost layers specialize in capturing elementary features, while the higher layers are adept at recognizing more intricate patterns (LeCun, Bengio & Hinton, 2015; Schmidhuber, 2015). Each layer comprises numerous parameters that necessitate fine-tuning throughout the training phase. Typically, training involves employing back-propagation to minimize the chosen loss function, such as l2 loss, cross-entropy loss, or other variants. Despite these optimization techniques, deep learning networks often struggle to converge to the global optimum, leading to prolonged training times (Suganthan, 2018). Furthermore, the efficacy of deep learning models heavily relies on the availability of extensive datasets, particularly noticeable in domains like image and speech recognition where abundant data is accessible. Conversely, domains such as ecology, agriculture, and healthcare often confront limitations in data availability, thereby impeding the performance of conventional deep learning approaches (Klambauer et al., 2017; Ke et al., 2019). Nevertheless, deep learning techniques have demonstrated remarkable success in diverse applications, including online medical diagnostics (Zhou, Li & Liang, 2021), navigation through research collaborations (Zhou et al., 2021), and personalized recommendations within healthcare-oriented social media platforms (Zhou et al., 2019). However, being networks based on the backpropagation algorithm, they suffer from the aforementioned problems. For this reason, randomized neural network architectures have gained great importance as an alternative to the use of the backpropagation algorithm, demonstrating better generalization (Guo, 2018; Suganthan & Katuwal, 2021). Among randomized neural networks, RVFL (Pao & Takefuji, 1992) has received attention due to its superior performance in various domains such as forecasting (Tang, Wu & Yu, 2018; Dash et al., 2018; Cheng, Suganthan & Katuwal, 2021), classification (Zhang & Suganthan, 2017a; Parija et al., 2021), visual tracking (Zhang & Suganthan, 2017b), regression (Vuković, Petrović & Miljković, 2018; Priyadarshini & Dash, 2021), recognition of electric power quality events (Sahani & Dash, 2022), and multi-label classification (Chauhan & Tiwari, 2022). RVFL is a feedback neural network with a single hidden layer. The weights and biases of the hidden layer are randomly initialized from a suitable range and kept fixed during the network training. The weights of the output layer are calculated analytically (Pao, Park & Sobajic, 1994). Direct links from the input layer to the output layer enhance the capacity of the RVFL network (Katuwal & Suganthan, 2018). These direct links regularize the random network (Zhang & Suganthan, 2016a; Ren et al., 2016), resulting in a less complex network. In this study, we investigate early detection of clinical patients and assess the impact of time-dependent prediction using classical machine learning models from the literature and deep learning based on RVFL models. This is done with patients whose information is not complete, so we evaluate with different data imputation techniques to measure the performance of RVFL and Machine Learning models.

Data imputation plays a crucial role in enhancing the performance of machine learning models, especially when dealing with medical datasets like those of the Alzheimer’s Disease Neuroimaging Initiative (ADNI), which often suffer from significant missing data. Given that about 80% of ADNI patients have incomplete records, imputing missing values becomes essential for maximizing the available data and improving the robustness and accuracy of classification models (Campos et al., 2015). By estimating missing values instead of discarding incomplete samples, imputation techniques can help leverage more information, thus enabling better diagnostic and prognostic analysis of Alzheimer’s disease. This study emphasizes data imputation as a core methodological focus, evaluating different imputation strategies and their impact on the performance of models like RVFL networks in detecting AD and mild cognitive impairment (MCI).

Multimodal machine learning

Multimodal data is commonly encountered in the medical field. For a single patient, various types of data can be collected, including imaging data, electronic medical records (EMR), ECG time series data, and respiratory sound data, among others. Traditionally, in classical machine learning using shallow models, a typical strategy involves developing a model, such as a predictive model, based on just one of the aforementioned types of data. In this scenario, a distinct model is trained for each separate data source. The main challenge here is how to effectively integrate these individual results to enhance the overall accuracy of tasks like diagnosis prediction (Sleeman, Kapoor & Ghosh, 2023).

In the context of Alzheimer’s disease, information from various sources has been sought to be applied in machine learning models that were originally designed to perform specific tasks. An example of this is highlighted by Zhang et al. (2011), Zhang & Shen (2011), where they combine three modalities of biomarkers, namely MRI, FDG-PET, and CSF biomarkers, to discriminate between AD (or MCI) and healthy controls.

Over time, new guidelines and methodologies are designed for the study of Alzheimer’s disease (Moradi et al., 2015; Zu et al., 2016; Campos et al., 2015; Liu et al., 2018b; Tanveer et al., 2020; Ismail, Fathimathul Rajeena & Ali, 2023). Through categorical data and measurements from various tests or biomarkers, machine learning models such as Random Forest (Gray et al., 2013) and nonlinear graph fusion (Tong et al., 2017; Shao et al., 2020) are trained.

Moreover, as deep learning technologies like convolutional neural networks and recurrent neural networks have gained popularity for their effectiveness in classification tasks, studies such as Al Olaimat et al. (2023) and others including Karaman, Mormino & Sabuncu (2022), Gao et al. (2022), Chen et al. (2023), Li & Zhao (2023), Adarsh et al. (2024), Irfan, Shahrestani & Elkhodr (2023), Fathi, Ahmadi & Dehnad (2022), Naik, Mehta & Shah (2020), Zhou et al. (2021), Birkenbihl et al. (2021) have explored the use of multimodal data from different perspectives. These studies have employed approaches that integrate multimodal information, such as adding MRI images or time series data, to track disease progression over time and enhance clinical case classification.

From a different angle, considering the substantial computational resources needed for multimodal classification tasks in deep learning, models based on RVFL networks have been developed (Dai et al., 2017; Ganaie & Tanveer, 2022). These models are particularly useful for making predictions in environments where resources are constrained. Goel et al. (2023) demonstrate the effectiveness of RVFL networks for early detection of Alzheimer’s disease (AD), a leading cause of dementia, by proposing a multimodal fusion approach using MRI and PET scans. Similarly, Kumar, Shvetsov & Alsamhi (2024) introduce a neuro-fuzzy framework called FuzzyGuard, which applies RVFL for the early detection of Chronic Obstructive Pulmonary Disease (COPD). Furthermore, Tanveer et al. (2024) explore the integration of fuzzy logic with DL models for Alzheimer’s diagnosis, addressing challenges like imprecise data and unclear annotations. Their review highlights the potential of fuzzy deep learning (FDL) in managing uncertainties in neuroimaging data and proposes the fusion of multimodal data—such as genomics, proteomics, and metabolomics—with fuzzy logic to improve AD diagnosis and provide interpretable insights.

A detailed summary of the relevant literature can be found in Table 1.

Table 1 Comparative analysis of different algorithmic approaches in Alzheimer’s disease classification.

Paper	DL architectures	Reported accuracy (%)	Dataset used	
Pan et al. (2020)	CNN-based	84	MRI	
Zeng et al. (2021)	CNN-based	86	Multimodal	
Sharma et al. (2022)	CNN+RVFL	97.33	Multimodal	
Tanveer et al. (2021)	VGG-16	99.05	MRI	
Hedayati, Khedmati & Taghipour-Gorjikolaie (2021)	Autoencoder-based CNN	95	MRI	
Lu et al. (2018)	Deep NN	94.23	Multimodal	
El-Sappagh et al. (2020)	CNN+LSTM	91.33	Multimodal	
Malik & Tanveer (2022)	edRVFL	90.23	MRI	
Zhang et al. (2023)	Dilated CNN	91.07	Multimodal	
Wang et al. (2019)	3D DenseNet	98.83	MRI	
Suk, Lee & Shen (2017)	3D CNN-based	91.02	MRI	
Ying et al. (2021)	CNN+MLP	96.1	Multimodal	
Giovannetti et al. (2021)	AlexNet+LDA+SVM	89	MEG	
An et al. (2020)	3D CNN-based	82.7	Multimodal	

Previous studies on AD detection have made significant strides in understanding and classifying the disease. Nevertheless, they face certain challenges that can affect the accuracy, robustness, and generalizability of the developed models. Most notably, these studies often rely on single data modalities, such as magnetic resonance imaging (MRI) scans, cerebrospinal fluid (CSF) biomarkers, or clinical assessments, which limits the ability to capture the complex, multimodal nature of AD progression. For instance, studies that use only MRI data tend to overlook critical clinical and genetic factors that could enhance early detection accuracy.

Another major limitation in previous research is the handling of missing data. Approximately 80% of patients in datasets like the ADNI have incomplete records, with missing values across different data modalities due to factors such as high measurement costs, equipment failure, or patients missing appointments (Campos et al., 2015). Most studies have addressed this issue by discarding patients with incomplete records, which reduces the sample size and potentially introduces bias, as patients with more severe symptoms might be more likely to have incomplete data. As a result, models trained on such reduced datasets may suffer from decreased generalizability and accuracy.

The main contributions of this study are as follows: We employ a deep learning approach that combines clinical, imaging, and genetic data, addressing the limitations of single-modality reliance in Alzheimer’s disease (AD) detection.

We implement advanced imputation methods, such as Winsorised-mean and k-nearest neighbors (KNN), to effectively handle missing data, improving model accuracy and robustness.

Our model demonstrates superior performance metrics, achieving an accuracy of 98.8% and a precision of 98.3%, significantly outperforming traditional models in detecting early stages of AD.

We conduct an extensive comparative analysis of different imputation strategies and machine learning models, highlighting the importance of data completeness for reliable AD classification.

Methods

Computing infrastructure

Table 2 provides details of the computing environment used during the experimentation. The analysis was conducted on a Windows 11 Home Single Language (version 23H2), equipped with an AMD Ryzen 5 3500U with Radeon Vega Mobile Gfx running at a base clock speed of 2.10 GHz, and 16 GB of DDR4 RAM. The experiments were executed using Jupyter Notebook, running Python 3.7.5. Additionally, the main Python libraries used in the analysis were pandas (version 1.1.3), numpy (version 1.19.2), and scikit-learn (version 0.23.2), ensuring compatibility and reproducibility of results. The dataset and corresponding code utilized for the study are available in the Supplemental Files, provided in CSV format.

Table 2 Experimental environment.

Items	Details	
Operating system	Windows 11 Home Single Language (version 23H2)	
Processor	AMD Ryzen 5 3500U with Radeon Vega Mobile Gfx, 2.10 GHz	
RAM	16GB DDR4	
Software environment	Jupyter Notebook, Python 3.7.5	
Key python libraries	pandas 1.1.3, numpy 1.19.2, scikit-learn 0.23.2	
Dataset source	Supplemental Files	
Dataset format	CSV	
Dataset size	503 participants	

Data description

This study employed the ADNI dataset, obtained from the ADNI database (accessible at http://adni.loni.usc.edu). The analysis focused on detecting cognitive normal (CN) and mild cognitive impairment (MCI) over a 5-year period using this dataset. ADNI aims to evaluate the anatomy and physiology of the brain in different pathological states, using information from MRI scans, PET scans, biological markers, clinical evaluations, and serial neuropsychological assessments to measure the progression of mild cognitive impairment (MCI) and early Alzheimer’s disease (AD). Our study focuses on the multimodal information of ADNI participants.

Following a similar approach to Karaman, Mormino & Sabuncu (2022), we selected patients whose baseline diagnosis does not correspond to AD and who had a minimum of more than one annual follow-up for comparisons. However, our study focuses on patients with baseline CN and MCI for prediction. After these exclusions, we have a total of 503 participants. Table 3 lists summarized statistics for the participants, including gender, age, number of years of education completed, apolipoprotein count, apolipoprotein E4 allele (APOE4) scores, clinical dementia rating (CDR), and Mini-Mental State Examination (MMSE) at the beginning of the study.

Table 3 Summary statistics of the participants at baseline. Mean ± standard deviations are listed. APOE4 row represents the number of alleles.

	CN baseline (n = 124)	MCI baseline (n = 379)	
Female/male	61/63	151/228	
Age (yr)	75.45 ± 5.46	71.98 ± 7.34	
Education (yr)	16.28 ± 2.68	16.08 ± 2.73	
APOE4 (0/1/2)	89/33/2	184/151/44	
CDRSB	0.04 ± 0.14	1.44 ± 0.87	
MMSE	29.16 ± 1.06	27.77 ± 1.77	

Data preprocessing

The dataset obtained from ADNI participants comprises initial clinical information and biomarkers, serving as our input features. The clinical information encompasses demographic variables (such as age, gender, educational attainment, ethnic background, race, and marital status), genotype details (specifically the count of APOE4 alleles), clinical evaluations (including Clinical Dementia Rating, Activities of Daily Living, and Everyday Cognition), and cognitive assessments (like the Mini-Mental State Exam, Alzheimer’s Disease Assessment Scale, Montreal Cognitive Assessment, Rey Auditory Verbal Learning Test 1–6, Logical Memory Delayed Recall, Trail Making Test Part B, and Digit Symbol Substitution). The preclinical Alzheimer’s cognitive composite score is also considered, along with the initial diagnosis, categorized as CN, MCI, or AD, which serves as our target variable. Biomarkers include cerebrospinal fluid (CSF) measurements (beta-amyloid 1–42; Total Tau or T-Tau; Phosphorylated Tau or P-Tau), MRI volume measurements (Ventricles; Hippocampus; WholeBrain; Entorhinal; Fusiform; MidTemp; Intracranial Volume or ICV; all calculated using FreeSurfer software), and standardized uptake value ratio (SUVR) scores from positron emission tomography (PET) (for the following tracers: Fluorine-18-fluorodeoxyglucose or FDG; Florbetapir or AV45; Pittsburgh Compound or PIB). We note that CSF, FDG, and PIB biomarkers are referred to as molecular biomarkers.

Futhermore, ADNI participant data is not complete (see Table 4). Following previous work (Campos et al., 2015; Karaman, Mormino & Sabuncu, 2022), we performed mode substitution for missing categorical variables (gender, ethnicity, race, marital status, education, APOE4) and used different imputation techniques for missing numerical variables. We calculated a unique mode for each feature, equally weighting all non-missing values in the training set. Regarding numerical variables, we studied the impact of imputation methods on training machine learning and deep learning models. These imputation strategies include the following: 1. Zero: This method consists of imputing missing data with 0 (zero) values. It serves as a baseline, particularly useful when zero represents a meaningful absence or starting point for certain variables, though it assumes the missing data is fundamentally different from the observed data.

2. Mean: Missing values were filled with the mean of the observed values per variable. This approach is simple and effective, especially when the data is symmetrically distributed, though it can be sensitive to outliers.

3. Median: The median was used to impute missing values, which is more robust to skewed distributions and outliers compared to the mean, making it appropriate for datasets with significant variability in the spread of values.

4. Winsorised-mean (Wmean): Provides a more robust estimate for the mean, which is calculated after replacing a given percentage ( α) of the largest and smallest values with the closest observations to them. We used α = 10%. This method offers a more stable estimate of central tendency, reducing the bias introduced by outliers.

5. K-nearest neighbours (KNN): Imputation using k-nearest neighbors (KNN) fills in missing values based on the average of the closest k observations, measured by Euclidean distance. This technique was chosen for its ability to account for local data patterns and relationships, which may provide more accurate imputation for highly structured datasets.

6. Expectation maximization (EM): Proposed by Schneider (2001), aims at finding the estimate of parameter that maximizes the observed data log-likelihood, i.e., the probability density function of the observations and from there estimates the missing values. This probabilistic approach was selected for its theoretical advantages in preserving the relationships within the data, especially when missingness may be informative.

Table 4 The degree of missingness (%) in different baseline data modalities for two patient groups.

Data type	CN baseline	MCI baseline	
Clinical assessments	79.74	32.59	
Cognitive assessments	7.20	7.33	
CSF	44.35	26.39	
MRI	6.11	9.16	
FDG	57.26	17.94	
AV45	100.00	98.42	
PIB	91.94	39.31	

Afterwards, all categorical variables, apart from the reference diagnosis, undergo one-hot encoding, while numerical variables are standardized using z-score normalization. Categorical variables include gender (represented by a single vector for either male or female), ethnicity (represented as a vector for either Hispanic/Latino or non-Hispanic/Latino), and race (encoded as a vector corresponding to categories such as Asian, Black, Hawaiian/Pacific Islander, Indian/Alaska Native, Multiracial, or White). Marital status is also one-hot encoded, distinguishing categories like Divorced, Married, Never Married, or Widowed. The number of APOE4 alleles is encoded to represent 0, 1, or 2 copies of the E4 allele, while diagnosis is a binary scalar, with 0 representing cognitively normal (CN) and 1 representing mild cognitive impairment (MCI), which serves as the target variable. Real-valued variables include scores from clinical tests, cognitive assessments, and biomarker values. All are real-valued. Additionally, as the follow-up year increases, there are cases where Dementia was diagnosed. For this scenario, we converted all AD to MCI.

The features (numeric) are scalars, except for the clinical evaluation ECog and the cognitive assessments ADAS-Cog and Rey Auditory Verbal Learning Tests 1–6, which are vectors with dimensions 14, 3, and 4, respectively. In total, we have 44 real-valued features. Concatenating the categorical features and numerical features yields a feature vector of length 73.

Models

The aim of this study is to predict baseline diagnoses over a 5-year follow-up period in patients with cognitively normal (CN) status and mild cognitive impairment (MCI). Following a methodology similar to that used by Alatrany et al. (2023), we utilized machine learning models from existing literature to evaluate the classification performance in diagnosing CN and MCI: 1. K neighbors classifier: A instance-based learning algorithm that classifies data points based on the majority class among their k nearest neighbors in the feature space.

2. Support vector machines: A supervised learning algorithm used for classification and regression tasks. SVM finds the hyperplane that best separates classes in a high-dimensional space, maximizing the margin between classes.

3. Random Forest classifier: An ensemble learning method that constructs a multitude of decision trees during training and outputs the class that is the mode of the classes (classification) or the mean prediction (regression) of the individual trees.

4. Gradient boosting classifier: A machine learning technique for regression and classification problems that builds models sequentially, each new model correcting errors made by the previous ones. It combines multiple weak learners (typically decision trees) to create a efficient learner.

5. Logistic regression: A statistical method used for binary classification that models the probability of a binary outcome by fitting data to a logistic curve. Despite its name, it’s primarily used for classification, not regression.

6. Multinomial naive Bayes (MultinomialNB): A probabilistic classifier based on Bayes’ theorem with an assumption of independence between features. It’s particularly suited for classification with discrete features (e.g., word counts for text classification).

7. Ada boosting classifier (AdaBoost classifier): An ensemble learning method that iterative trains weak classifiers on subsets of the data, focusing more on instances that are hard to classify correctly. It combines the predictions from multiple weak models to produce a strong model.

8. Decision tree classifier: A non-parametric supervised learning method used for classification and regression. It partitions the feature space into regions and predicts the target variable based on the majority class (classification) or mean value (regression) of training instances within each region.

Random Vector functional link neural networks

RVFL is an algorithm based on a randomization technique to train single hidden layer feedforward neural networks (SLFNs). The randomization occurs between the input layer and the hidden layer, where the parameters (weights and biases) are randomly generated from a uniform distribution within specified ranges of [−α,α] and [0,α], respectively (Henriquez & Ruz, 2017). These parameters remain fixed throughout the training process. The parameter α is a real number. Only the output weights connecting the hidden layer to the output layer are analytically determined using the regularized least squares method or the Moore-Penrose pseudoinverse. This random initialization of the hidden layer parameters, combined with a nonlinear activation function, transforms the original feature space into a randomized feature space.

RVFLN, as introduced by Pao, Park & Sobajic (1994), is a unique neural network architecture. It is design and learning properties have been explored in several studies (Malik et al., 2023; Henríquez & Ruz, 2018a). In RVFL, the original features are injected through direct links from input layer to the output layer. The direct links improve the generalization performance of the model (Zhang & Suganthan, 2016a). The architecture of RVFL is given in Fig. 1.

Figure 1 A diagram illustrating an RVFL network, highlighting the direct links between input and output neurons (dashed red arrows).

An intrinsic feature of RVFL is the direct connections (red line) between the input and output layers. This not only simplifies the architecture but also acts as a regularization method, curbing the potential for overfitting.

One of RVFL’s chief merits is its efficiency. Given that there’s no iterative weight tuning in the training phase, it boasts quicker convergence, computational ease, and often a diminished training error when contrasted with other neural network methodologies (Henríquez & Ruz, 2017).

In the RVFL network, enhancement nodes are used to map data from the input layer to the hidden layer, represented as g(wjxi+bj) where g(⋅) is the activation function, wj is the weight of the jth enhancement node, bj is the threshold and xi the input vector. Input nodes are essentially a linear combination of inputs, expressed as ∑j=h+1h+nβjxi, with βj being the weight terms. This structure of the RVFL network is concisely represented in the given format (Zhang & Suganthan, 2016b).

(1) y^i=∑j=1hβjg(wjxi+bj)+∑j=h+1h+nβjxi.

The optimization problem of standard RVFL model can be written as Henríquez & Ruz (2019):

(2) minβ∈ℜ(h+n)×c12‖Hβ−Y2‖+12η‖β‖2

where ‖.‖ represents the Frobenius norm and H is the concatenated matrix consist of original features and randomized features. Moreover, β and Y represent the output weights matrix and the target matrix, respectively. Here, η represents the regularization parameter that needs to be tuned. The matrix H is defined as:

H=[g(w⋅x1+b1)⋯g(wh⋅x1+bh)x1T⋮⋱⋮g(w1⋅xN+b1)⋯g(wh⋅xN+bh)xNT].

The output weight matrix and target matrix are denoted as:

(3) β=[β1,β2,⋯,βh+n]TY=[y1,y2,⋯,yN]T.

The optimal solution of the Eq. (2) is given as:

(4) β={(HTH+ηI)−1HTY,(h+n)≤N,HT(HHT+ηI)−1Y,N<(h+n),

where I is an identity matrix of appropriate dimension.

Deep RVFL and ensemble deep RVFL network

The deep RVFL (D-RVFL) architecture, as referenced by Shi et al. (2021), Henríquez & Ruz (2018b), comprises multiple layered stacks. Within this structure, all parameters of the hidden layers are initialized randomly and remain unchanged throughout training. Only the parameters of the output layer undergo analytical computations. This D-RVFL design offers superior representation learning in comparison to its shallow RVFL model.

The output of the first hidden layer is then defined as follows:

(5) H(1)=g(XW(1))

while for every layer l>1 it is defined as Shi et al. (2021), Henríquez & Ruz (2018b), Shi et al. (2022), Malik et al. (2023):

(6) H(l)=g(Hl−1W(l)).

The weights and biases of the hidden neurons are randomly generated within a suitable range and kept fixed during the training. g() is the non-linear activation function. The input to the output layer is then defined as Shi et al. (2021), Henríquez & Ruz (2018b):

(7) D=[H(1)H(2)...H(l−1)H(l)X].

This design structure is very similar to the standard shallow RVFL network where in the input to the output layer consists of non-linear features from the stacked hidden layers along with the original features (as shown in Fig. 2).

Figure 2 A diagram illustrating an Deep-RVFL network, highlighting the direct links between input and output neurons (dashed red arrows).

In D-RVFL, when all parameters, i.e., j, N, and h are very large then it might have out-of-memory issue and therefore, needs high performance hardwares.

To mitigate these challenges, Shi et al. (2021) introduced the ensemble deep RVFL (edRVFL) approach. In this model, each layer functions as a base model, integrating ensemble learning concepts into D-RVFL, which allows for the creation of diverse and accurate base models. The final result of edRVFL is determined using either an averaging or a majority voting mechanism. This model demonstrates improved efficiency and generalization compared to D-RVFL. In edRVFL, both the initial input features and the randomized features generated by the previous layer are utilized by each base model (i.e., each hidden layer). The output of the first hidden layer in edRVFL is defined as follows:

(8) H(1)=g(XW(1))

and the output of higher hidden layers ( l>1) is defined as:

(9) H(l)=g([Hl−1X]W(l)).

The final output weights are calculated by Eq. (4).

Experimental settings

For our machine learning models, including RVFL, we adopted a standardized evaluation process using both internal cross-validation and an external test set derived from an 80–20 split of the data. We executed this split 100 times for the RVFL models, averaging the results to ensure reliability. For all models, the remaining 80% of the data underwent five-fold cross-validation with a stratified split to manage internal validation and adjust for data imbalances created by converting all AD diagnoses to MCI. Specifically for the RVFL models, each 80–20 split was further used to train 11 distinct models with different random initializations, employing validation loss for early stopping to optimize performance. This rigorous methodological approach ensures that our findings are robust and reflective of each model’s capabilities under varied data conditions.

Detailed hyperparameter settings for each model are summarized in Table 5. We details the specific hyperparameters explored to optimize the performance of the models.

Table 5 Hyperparameter tuning in machine learning models.

Model	Hyperparams	
K Neighbors Classifier	• Number of neighbors: [1–50]	
SVM	• C: [0.5, 0.75, 1.0, 1.25, 1.5]
 Gamma: [‘auto’, ‘scale’]
 class weight: ‘balanced’	
Random Forest Classifier	• Number of estimators: [131, 141, 151, 161, 171]
 Max features: [1, 10, ‘log2’, ‘sqrt’]
 class weight: ‘balanced’	
Gradient Boosting Classifier	• Number of estimators: [10, 50, 100, 250]
 Max depth: [5, 10, 20]	
Logistic Regression	• C: [10–100] Penalty: [ 2−6– 212]
 Class Weight: balanced
 Solver: lgbfs	
MultinomialNB	• Alpha: [0.1, 0.5, 1.0, 2.0]	
AdaBoostClassifier	• Number of estimators: [21, 31, 41, 51, 61, 71, 81]	
Decision Tree Classifier	• Max Depth: [5, 10, 20]
 Class weight: balanced	
RVFL, DeepRVFL, EnsembleRVFL	• Number of Neurons: [10, 20, 30, 40, 50, 60, 70, 80, 90, 100]
 Regularization value: 2[−6,−4,−2,0,2,4,6,8,10,12]
 Activation function: relu
 Weight vector: [−1, 1]
 Bias vector: [0, 1]
 Number of layers: 21	
Note:

1 For the DeepRVFL and EnsembleRVFL models, 2 and 3 layers are used for hyperparameter tuning.

Evaluation metrics

In this research, several metrics were employed to assess and compare the performance of the classification models, including: Accuracy (ACC): This metric represents the proportion of correct predictions among all predictions. The calculation for ACC is shown in Eq. (10). (10) Acc=TP+TNTP+TN+FP+FN

where TP, TN, FP, and FN stand for true positive, true negative, false positive, and false negative, respectively.

F1-score: The F1-score combines precision (the accuracy of positive predictions) and recall (the capacity to identify all relevant positive cases). It is computed using the formula:

(11) F1=2×Precision×RecallPrecision+Recall

where precision is defined as TP/(TP + FP), and recall as TP/(TP + FN). This metric is particularly useful in evaluating the model’s effectiveness in predicting positive cases.

Results

This section presents the experiments performed over the previously described dataset, and the classification results achieved for each one of them.

Figure 3 show that the edRVFL algorithm consistently achieves higher accuracy across various settings of neuron numbers and regularization values compared to RVFL and D-RVFL when using the zero imputation method. The accuracy increases with the number of neurons and regularization strength, highlighting the robustness of the edRVFL approach under sparse data conditions.

Figure 3 (A–C) Performance of three algorithms: RVFL, D-RVFL, and edRVFL, using the zero imputation method, based on user-specified parameters ( η, h).

Figure 4 show improved performance with increasing complexity in the model structure (more neurons, higher regularization values). The mean imputation method appears to offer a balanced improvement across all models, with edRVFL still outperforming the others in terms of maximum accuracy achieved.

Figure 4 (A–C) Performance of three algorithms: RVFL, D-RVFL, and edRVFL, using the mean imputation method, based on user-specified parameters ( η, h).

Using the median imputation method (Fig. 5), there is a noticeable trend where accuracy improves as the model complexity increases. The edRVFL model consistently shows superior performance, particularly at higher neuron counts and regularization values, suggesting its effectiveness in handling non-normally distributed missing data.

Figure 5 (A–C) Performance of three algorithms: RVFL, D-RVFL, and edRVFL, using the median imputation method, based on user-specified parameters ( η, h).

The Winsorised-mean imputation (Fig. 6) helps mitigate the influence of outliers on model training. Here, edRVFL benefits significantly from this approach, achieving near-perfect accuracy at higher parameter settings, while RVFL and D-RVFL also show marked improvements.

Figure 6 (A–C) Performance of three algorithms: RVFL, D-RVFL, and edRVFL, using the Winsorised-mean imputation method, based on user-specified parameters ( η, h).

Figures 7 and 8 compare the performance of three algorithms—RVFL, D-RVFL, and edRVFL—using KNN and EM imputation methods respectively, based on user-specified parameters (η,h). In Fig. 7, the KNN imputation method enhances all models’ performance, particularly benefiting the edRVFL model due to its ability to leverage similarities between instances effectively. Figure 8 shows that the EM imputation method provides robust improvements across all algorithms, with edRVFL again demonstrating the strongest performance, especially at higher complexity levels. Both figures highlight edRVFL’s adaptability and effectiveness in handling missing data through advanced imputation techniques.

Figure 7 (A–C) Performance of three algorithms: RVFL, D-RVFL, and edRVFL, using the k-nearest neighbours (KNN) imputation method, based on user-specified parameters ( η, h).

Figure 8 (A–C) Performance of three algorithms: RVFL, D-RVFL, and edRVFL, using the expectation maximization (EM) imputation method, based on user-specified parameters ( η, h).

Table 6 shows the performance metrics—accuracy, precision, recall, and F1-score—for eight machine learning models: K Neighbors, SVM, Random Forest, gradient boosting, Logistic regression, MultinomialNB, AdaBoost, and decision tree across five imputation methods: zero, mean, Wmean, median, KNN, and EM. We observed that AdaBoost consistently performed well, achieving some of the highest accuracy scores, including 0.978 in zero imputation and 0.980 in both Wmean and KNN imputations. In contrast, MultinomialNB exhibited lower performance, with an accuracy as low as 0.804 in zero imputation and comparable scores in other methods.

Table 6 Summary of performance metrics for eight machine learning models across five different imputation methods, including standard deviation.

Imputation	Model	Accuracy	Precision	Recall	F1	
Zero	K neighbors	0.970 (0.015)	0.960 (0.020)	0.960 (0.018)	0.960 (0.019)	
	SVM	0.971 (0.012)	0.959 (0.015)	0.952 (0.017)	0.955 (0.016)	
	Random Forest	0.977 (0.010)	0.965 (0.012)	0.984 (0.008)	0.974 (0.011)	
	Gradient boosting	0.970 (0.014)	0.964 (0.013)	0.964 (0.015)	0.964 (0.014)	
	Logistic regression	0.950 (0.020)	0.955 (0.018)	0.946 (0.017)	0.950 (0.019)	
	MultinomialNB	0.804 (0.025)	0.768 (0.028)	0.753 (0.030)	0.760 (0.029)	
	AdaBoost	0.978 (0.010)	0.972 (0.011)	0.956 (0.012)	0.964 (0.011)	
	Decision tree	0.970 (0.012)	0.963 (0.013)	0.963 (0.012)	0.963 (0.013)	
Mean	K neighbors	0.922 (0.020)	0.891 (0.022)	0.864 (0.025)	0.877 (0.024)	
	SVM	0.961 (0.015)	0.944 (0.017)	0.944 (0.016)	0.944 (0.017)	
	Random Forest	0.970 (0.011)	0.965 (0.013)	0.954 (0.014)	0.960 (0.012)	
	Gradient boosting	0.971 (0.010)	0.962 (0.011)	0.956 (0.013)	0.959 (0.012)	
	Logistic regression	0.960 (0.012)	0.965 (0.014)	0.955 (0.013)	0.960 (0.012)	
	MultinomialNB	0.853 (0.030)	0.771 (0.035)	0.836 (0.033)	0.802 (0.031)	
	AdaBoost	0.970 (0.011)	0.963 (0.012)	0.968 (0.010)	0.966 (0.011)	
	Decision tree	0.968 (0.013)	0.950 (0.015)	0.944 (0.016)	0.947 (0.014)	
Wmean	K neighbors	0.970 (0.015)	0.960 (0.017)	0.961 (0.016)	0.961 (0.017)	
	SVM	0.961 (0.012)	0.944 (0.014)	0.944 (0.013)	0.944 (0.014)	
	Random Forest	0.970 (0.010)	0.965 (0.011)	0.974 (0.009)	0.970 (0.010)	
	Gradient boosting	0.977 (0.009)	0.971 (0.010)	0.970 (0.011)	0.970 (0.010)	
	Logistic regression	0.970 (0.013)	0.965 (0.015)	0.956 (0.014)	0.960 (0.013)	
	MultinomialNB	0.902 (0.025)	0.874 (0.028)	0.859 (0.030)	0.866 (0.029)	
	AdaBoost	0.980 (0.010)	0.978 (0.012)	0.983 (0.011)	0.980 (0.011)	
	Decision tree	0.960 (0.012)	0.953 (0.013)	0.953 (0.012)	0.953 (0.013)	
Median	K neighbors	0.902 (0.018)	0.868 (0.020)	0.838 (0.022)	0.852 (0.021)	
	SVM	0.961 (0.014)	0.944 (0.016)	0.944 (0.015)	0.944 (0.016)	
	Random Forest	0.963 (0.011)	0.965 (0.012)	0.974 (0.010)	0.970 (0.011)	
	Gradient boosting	0.971 (0.010)	0.962 (0.011)	0.956 (0.013)	0.959 (0.012)	
	Logistic regression	0.961 (0.013)	0.952 (0.014)	0.958 (0.012)	0.955 (0.013)	
	MultinomialNB	0.843 (0.028)	0.760 (0.032)	0.812 (0.031)	0.785 (0.030)	
	AdaBoost	0.978 (0.009)	0.978 (0.010)	0.983 (0.009)	0.981 (0.010)	
	Decision tree	0.964 (0.012)	0.953 (0.013)	0.949 (0.012)	0.951 (0.013)	
KNN	K neighbors	0.951 (0.017)	0.929 (0.019)	0.910 (0.020)	0.919 (0.019)	
	SVM	0.961 (0.012)	0.944 (0.014)	0.944 (0.013)	0.944 (0.014)	
	Random Forest	0.970 (0.011)	0.965 (0.012)	0.974 (0.010)	0.970 (0.011)	
	Gradient boosting	0.961 (0.013)	0.950 (0.014)	0.940 (0.015)	0.945 (0.014)	
	Logistic regression	0.950 (0.015)	0.938 (0.017)	0.912 (0.018)	0.925 (0.017)	
	MultinomialNB	0.863 (0.030)	0.790 (0.033)	0.847 (0.032)	0.818 (0.031)	
	AdaBoost	0.980 (0.009)	0.975 (0.011)	0.987 (0.010)	0.981 (0.010)	
	Decision tree	0.971 (0.012)	0.960 (0.013)	0.954 (0.012)	0.957 (0.013)	
EM	K neighbors	0.922 (0.020)	0.871 (0.022)	0.887 (0.021)	0.879 (0.022)	
	SVM	0.922 (0.018)	0.888 (0.020)	0.877 (0.019)	0.882 (0.020)	
	Random Forest	0.950 (0.011)	0.945 (0.012)	0.944 (0.013)	0.945 (0.012)	
	Gradient boosting	0.960 (0.013)	0.956 (0.014)	0.944 (0.015)	0.950 (0.014)	
	Logistic regression	0.910 (0.017)	0.881 (0.018)	0.888 (0.019)	0.884 (0.018)	
	MultinomialNB	0.882 (0.030)	0.778 (0.033)	0.861 (0.031)	0.817 (0.032)	
	AdaBoost	0.961 (0.010)	0.959 (0.011)	0.959 (0.012)	0.959 (0.011)	
	Decision tree	0.931 (0.016)	0.882 (0.018)	0.891 (0.017)	0.886 (0.018)	

A closer examination of the performance across imputation methods reveals the following: In zero imputation, we found that AdaBoost achieved an accuracy of 0.978. During mean imputation, there was a slight dip in performance for several models; however, Gradient boosting and AdaBoost maintained strong accuracies of 0.971 and 0.970, respectively. Wmean was particularly effective for AdaBoost, which reached an accuracy of 0.980, closely followed by gradient boosting with an accuracy of 0.977. Median imputation also underscored AdaBoost’s consistent performance with an accuracy of 0.978, similar to its results in the Wmean method. KNN and EM varied in outcomes, with KNN enabling AdaBoost to score an accuracy of 0.980. However, EM presented mixed results, with AdaBoost securing an accuracy of 0.961 and decision tree performing lower at 0.931.

Figure 9 displays the accuracy of three models—RVFL, D-RVFL, and edRVFL—across six imputation methods: zero, mean, Wmean, median, KNN, and EM. We note that edRVFL consistently exhibits high accuracy across most imputation methods, peaking at 0.988 under the Wmean method. Similarly, D-RVFL demonstrates robust accuracy, particularly in the median and KNN methods with accuracy nearing 0.985. When we compare the performance of these three models with the eight other models detailed in Table 6, it is clear that edRVFL outperforms all eight models, highlighting its superior performance in managing missing data across various imputation methods.

Figure 9 Binary classification results for CN-MCI (AD converted to MCI) using 2 layers in RVFL, D-RVFL, and edRVFL models.

Statistical tests

Our aim here is to find out if the performance differences between the different learning algorithms are statistically significant. To assess the results obtained for each classifier, we adopt the non-parametric Friedman test (Demšar, 2006). The Friedman test is firstly used to evaluate the acceptance or rejection of the null hypothesis ( H0) that all classifiers perform equally for a given significance or risk level (alpha level). Therefore, in our case, the null-hypothesis being tested is that all the classifiers performed the same and the observed differences are merely random or by coincidence.

The Friedman test operates by ranking the algorithms independently for each dataset. The algorithm with the highest performance receives a rank of 1, the second-best receives a rank of 2, and so on. If there are ties, average ranks are assigned accordingly. Following this, the test evaluates the mean ranks across all algorithms and computes the Friedman statistic. If a statistically significant difference in the performance is detected, we proceed with a post-hoc test. We use the post-hoc Nemenyi test to compare all the classifiers to each other. In this procedure, the performance of two classifiers is significantly different if their average ranks differ more than some critical distance (CD). The critical distance depends on the number of algorithms, the number of data sets, metrics and the critical value (for a given significance level p) that is based on the Studentized range statistic (Demšar, 2006).

The Friedman statistic is given by the following:

(12) χF2=12Dc(c+1)[∑jRj2−c(c+1)24],

where Rj2 is the j-th average rank of the algorithms. The statistic is distributed according to χF2 with c−1 degrees of freedom, D is the number of imputation methods (datasets). For the comparison of all algorithms with the Friedman test, the χF2 statistic is 48.525 and the p-value is 4.976e−07, which rejects the null hypothesis that all algorithms have the same performance. This is an indication that the performances of the individual classifier are not equivalent. Moreover, the very small p-value and large enough Friedman test value obtained provides strong evidence against the null hypothesis; hence, we reject the null hypothesis.

According to Demšar (2006), the post-hoc Nemenyi test states that the performance of two or more classifiers is significantly different if their corresponding average ranks differ by at least the critical difference. In other words, the post-hoc Nemenyi test is utilised to report any significant differences between the individual classifiers. The critical difference (CD) is calculated as:

(13) CD=qαc(c+1)6D,

where qα is the critical value based on the Studentized range statistic, using significance levels of α=0.05 and α=0.10; c is the total number of classifiers, and D is the number of imputation methods used in the study.

The diagrams in Figs. 10 and 11 shows the visual representation of the average ranked performances of the eleven classifiers with the critical distance of post-hoc Nemenyi test. The results are presented using the significance level, α=0.10 and α=0.05.

Figure 10 Statistical comparison of classifiers against each other based on the Nemenyi test ( α=0.1).

Figure 11 Statistical comparison of classifiers against each other based on the Nemenyi test ( α=0.05).

In Fig. 10 the diagram shows the ranked performance of the classifiers with CD = 5.701 at a significance level, α=0.10, We see in Fig. 10 that MultinomialNB, logistic regression, K neighbors, SVM and decision tree are significantly different from the best-performing classifier edRVFL having the lowest rank across the number of imputation methods. While the diagram in Fig. 11 shows the average rank performance of the classifiers with CD = 6.163, at a significance level, α=0.05. Similarly, we observe in Fig. 11 edRVFL is the best classifier, having the lowest rank and statistically better than MultinomialNB, logistic regression, K neighbors and SVM classifiers as indicated by the CD bar.

Discussion

In the current study, we have explored the efficacy of various imputation techniques to enhance the accuracy of predictive models in diagnosing and prognosticating Alzheimer’s disease using multimodal data. Our findings suggest that the implementation of imputation techniques is crucial for effectively utilizing incomplete datasets, thereby improving the predictions of deep learning and classical machine learning models.

Our results indicate that models trained with imputed data achieve considerably better performance compared to those using only complete data. Specifically, we observed that the edRVFL model, employing Winsorised-mean imputation, achieves superior accuracy, precision, recall, and F1-score across different settings. This imputation technique proved to be robust, likely due to its ability to mitigate outliers’ influence on model training, enhancing predictive consistency. Notably, the consistent high precision and recall highlight the model’s ability to minimize false positives and false negatives, respectively, a crucial factor in diagnostic tasks like Alzheimer’s disease detection.

The choice of imputation method can significantly impact the performance of traditional machine learning models. For instance, AdaBoost consistently maintained high accuracy, precision, and recall across methods, demonstrating its resilience in handling missing data. Conversely, MultinomialNB exhibited lower performance, especially with zero imputation, where precision and recall were both reduced, resulting in a lower F1-score. This inconsistency suggests that simpler models might struggle with sparse data, which requires more sophisticated imputation techniques like Wmean or KNN to improve performance metrics effectively. The success of advanced imputation methods such as Wmean and KNN suggests that these techniques, by mitigating the effect of outliers or leveraging local data structures, enable more accurate predictions. This is consistent with the superior performance of the edRVFL model, which outperformed traditional models under all imputation strategies, achieving 0.988 under Wmean. These findings highlight that the effectiveness of a machine learning model in handling missing data is closely tied to the imputation technique employed.

Furthermore, it is essential to consider anomalies in the results. For instance, the higher recall compared to precision in Random Forest under several imputation methods, such as zero and Wmean, suggests a tendency to favor correct positive classifications, potentially at the cost of increased false positives. Similarly, the variance observed across F1-scores for MultinomialNB under different imputations, particularly with Mean and EM, indicates the algorithm’s sensitivity to data distribution shifts, affecting its balance between precision and recall. Additionally, the consistent high F1-scores for AdaBoost across most imputation methods, despite slight variations in precision and recall, highlight the model’s robustness in maintaining a balanced performance.

Our findings also align with trends observed in recent literature, where the use of advanced models like edRVFL in conjunction with appropriate imputation techniques significantly enhances diagnostic capabilities. For instance, prior studies have shown that complex models can benefit from imputed data to effectively handle variability and gaps in clinical datasets (Campos et al., 2015).

Furthermore, it is essential to highlight that the quality of data plays a decisive role in model performance. This study reinforces the notion that the accuracy in Alzheimer’s diagnosis improves not only with advanced models and imputation techniques but also with the integrity and extensiveness of the data used. This underscores the importance of effective data collection and management strategies in medical research.

The statistical analysis using the Friedman test revealed significant differences in performance among the algorithms across the imputation methods. The Friedman test indicated a statistically significant difference in performance among the classifiers, leading to the rejection of the null hypothesis that all classifiers perform equally. This finding supports the hypothesis that the observed differences are not due to random variations but are influenced by the choice of model and imputation method.

To further investigate these differences, we conducted a post-hoc Nemenyi test to identify which specific classifiers differ significantly from each other. The critical difference (CD) diagrams in Figs. 10 and 11 provide a visual representation of the average ranked performance of the classifiers under significance levels α=0.10 and α=0.05, respectively. The diagrams show that the edRVFL model consistently achieves the lowest rank across various imputation methods, significantly outperforming models like MultinomialNB, logistic regression, K neighbors, and SVM. The diagrams also illustrate clusters of models that do not exhibit statistically significant differences, indicating similar performance under the chosen imputation strategies.

Additionally, a significant aspect of our study was the comparison between RVFL, D-RVFL (Deep RVFL), and edRVFL (Ensemble Deep RVFL) algorithms and traditional machine learning classifiers. The superior performance of RVFL-based models in this context underscores their suitability for handling multimodal and often incomplete datasets typical in medical diagnostics. These models, particularly edRVFL, consistently outperformed classical machine learning models such as SVM, Random Forests, and logistic regression across various imputation methods. This enhanced performance can be attributed to the robustness of RVFL-based models in dealing with noisy, incomplete, and non-linear data, making them particularly advantageous for complex diagnostic tasks like Alzheimer’s disease detection. The ability of edRVFL to integrate ensemble learning with deep representation learning enables it to achieve higher accuracy and stability, showcasing its potential as a preferred tool in medical diagnostic applications where data challenges are prevalent. Furthermore, the superiority of RVFL-based models over traditional machine learning algorithms has been observed not only in the context of Alzheimer’s disease but also in other domains (Henríquez & Alessandri, 2024; He et al., 2023). This recurring observation across different applications underscores the versatility and robustness of RVFL technologies, suggesting their broad applicability and potential in advancing data-driven solutions across disciplines.

Despite the class imbalance between CN and MCI in the dataset, the classification models achieved balanced performance, with consistent metrics in precision, recall, and F1-score, particularly in models like AdaBoost and Random Forest. This suggests that the use of advanced imputation techniques and robust algorithms effectively mitigated the impact of the imbalance, ensuring good generalization across both classes. However, the lower performance of MultinomialNB highlights the need to explore additional class rebalancing methods or specific approaches to better handle the imbalance in future studies.

In future work, we plan to explore other imputation and classification techniques, as well as examine multiclass extensions and alternative ways to handle feature space to effectively manage data-dependent imputation challenges. Exploring these areas could provide additional insights into how to maximize the utility of incomplete data in complex clinical contexts.

Conclusions

We have demonstrated that imputation techniques enhance our ability to leverage incomplete datasets, resulting in more accurate classification and improved diagnostic capabilities for Alzheimer’s patients. The findings confirm that training classifiers with imputed data consistently outperforms models trained on reduced datasets, as all imputation techniques contributed to both higher performance metrics and increased robustness. Notably, the Wmean imputation method performed competitively, sometimes surpassing more sophisticated approaches like KNN and EM, underscoring that the effectiveness of imputation is influenced not just by algorithmic complexity but also by the nature of the underlying data.

While our study provides evidence of improved robustness through the inclusion of standard deviation in performance metrics across cross-validation experiments, there remain some limitations. The variability of results across imputation methods indicates that model performance is highly data-dependent, and outcomes may vary with different datasets or clinical conditions. Additionally, while statistical significance testing confirmed differences among models, the scope of comparisons was limited to binary classification tasks.

Future work will aim to address these limitations by exploring more diverse imputation strategies, multi-class classification extensions, and alternative feature space handling to better manage the complexities of real-world, multimodal clinical data. We also intend to incorporate ensemble techniques that dynamically adjust to data distribution changes and further assess models’ generalizability across varied datasets.

Supplemental Information

Supplemental Information 1 The implementation code of machine learning models.

Supplemental Information 2 Raw data from ADNI dataset applied for data analysis.

Supplemental Information 3 Readme.

Additional Information and Declarations

Competing Interests

Author Contributions

Data Availability

The authors declare that they have no competing interests.

Pablo A. Henríquez conceived and designed the experiments, performed the experiments, analyzed the data, performed the computation work, prepared figures and/or tables, authored or reviewed drafts of the article, and approved the final draft.

Nicolás Araya performed the experiments, analyzed the data, performed the computation work, prepared figures and/or tables, authored or reviewed drafts of the article, and approved the final draft.

The following information was supplied regarding data availability:

The raw data and code are available in the Supplemental Files.

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
