# Peer review of "Multimodal Alzheimer’s disease classification through ensemble deep random vector functional link neural network"

_PeerJ Computer Science, doi:10.7717/peerj-cs.2590_

## Round 0.1 · original submission · Major Revisions

The Authors exploit a multimodal dataset of AD to propose different DL and ML approaches, as well as imputation methods, to classify cognitive normal and mild cognitive impaired conditions.

While the topic is worth exploring and the paper is generally well written, the Reviewers highlight important concerns to be addressed.

Please, answer all the Reviewers’ concerns with a point-by-point response and in particular:

- Highlight the research questions/novel contributions of your work.
- Provide a clear assessment of the importance of using data imputation methods and why you use specific techniques.
- Consider adding up to date literature works. Reviewer 1 suggested a paper that you could use to search further references. Adding the suggested paper is optional and should be done only if you agree with the fact that it can improve the presentation of your work.
- Provide a clear assessment and discussion of the results. In particular, revise the values reported for each metric considering both Reviewer 1 and Reviewer 3 detailed comments on the matter. Moreover, a clear statement should be made regarding the dataset imbalance and how this problem is addressed.
- A statistical significance test between models would enhance the reliability of the reported results.
- Consider improving the reproducibility of your work by providing sufficient details on the performed computations.

Reviewer 1 ·

Basic reporting

The abstract must emphasize the research gap in Alzheimer's Disease (AD) detection and the key results obtained from the study. Additionally, explaining performance metrics (e.g., accuracy, precision) without the obtained results is insufficient.

Despite the importance of data imputation being highlighted in the discussion and experiment sections, there is no mention of it in the abstract or introduction. Data imputation often plays a crucial role in improving machine learning performance, especially in medical datasets. This should be mentioned early in the paper to reflect the study's methodological focus.

The related work section fails to clearly discuss the limitations of previous studies on AD detection. It’s crucial to explicitly mention these shortcomings and, more importantly, how the current study has addressed or overcome them. This would strengthen the justification for the research and support the claims made about the model's effectiveness.

The contributions of the study should be listed as clear, concise bullet points to ensure they are easily identifiable. This approach improves readability and helps the audience quickly understand the primary contributions of the work.

The novelty of the study is unclear and poorly presented. To better highlight the innovative aspects, the paper needs to clearly state what sets this work apart from existing methods in AD detection, whether it’s in the model, data processing techniques, or evaluation strategy.


The references in the article appear outdated. Alzheimer's Disease detection is an actively evolving field, and more recent works from the past two to three years should be cited to ensure the study remains relevant and engages with the latest developments.

A meta-heuristic multi-objective optimization method for Alzheimer’s disease detection based on multi-modal data
Multforad: Multimodal mri neuroimaging for alzheimer’s disease detection based on a 3d convolution model


Experimental design

Missing Statistical Analysis & Imbalanced Data Handling

The experiment section lacks statistical measures that provide insight into the significance of the results.

Additionally, the paper does not account for the effect of imbalanced data, which is common in medical datasets like those used for AD detection. Ignoring these factors can lead to misleading results.

The study should also consider the generalization and robustness of the model, as it’s important to ensure it works well across diverse populations.

The selected features for Alzheimer's Disease detection have not been optimized or adjusted based on their relevance to the task. It’s important to justify why certain features were chosen and whether they significantly contribute to the model’s performance.

Validity of the findings

There is a serious flaw in the classification accuracy and other key metrics such as precision and recall. Accuracy must be a weighted average of precision and recall. This inconsistency should be addressed, as it could indicate flaws in how the model handles data or other issues in the experimental setup.


The discussion lacks depth, particularly in terms of supporting the obtained metric values. Each metric (accuracy, precision, recall, F1 score) should be critically analyzed, and the authors must provide a solid rationale for the results, especially when anomalies or inconsistencies arise.

Annotated reviews are not available for download in order to protect the identity of reviewers who chose to remain anonymous.

·

Basic reporting

The usage of language in expressing the research idea is quite interesting.
No comments.

Experimental design

Experimental design and the proceedings seems to be well organised

Validity of the findings

RVFL_PROJECT_EDA.ipynb has a set of results that seems to be different from the one reported in Table 6.
Kindly ensure if the correct data is provided in the table.
If the results of the proposed method are measured towards ROC, training , and testing accuracy since deep networks are used.

Reviewer 3 ·

Basic reporting

The manuscript is well-organized and generally well-written, but the separation of the Model and Method sections reduces readability and introduces unnecessary complexity. In typical computer science papers, models are discussed within the Methods section, providing a cohesive view of the experimental process. Merging these sections would enhance the logical flow of the paper. Additionally, some technical explanations, particularly around the RVFL model, could be made more concise to improve accessibility for a broader audience. The paper is adequately referenced, but adding recent studies on RVFL in medical data contexts could further strengthen the literature review.

Experimental design

· The methods are well-described, but more information on the computing environment (e.g., hardware specifications, software versions) would enhance reproducibility.
· The discussion on data preprocessing, especially the handling of missing data, is sufficient, but further justification for the chosen imputation techniques (e.g., KNN, Winsorization) would be useful. A discussion on how these methods impact model performance would also add value.
· A key concern is the lack of reporting on variance or standard deviation across cross-validation experiments. Including these statistics would provide insight into the robustness of the results.
· Additionally, performing statistical significance tests between model performances would further support the claims made in the paper and ensure that observed differences are not due to random variability.
· While the evaluation metrics are clearly explained, reporting variance would better support the conclusions drawn about model performance.

Validity of the findings

The conclusions are well-aligned with the results presented in the paper, but the lack of variance reporting for the cross-validation experiments raises concerns about the robustness of the findings. Including standard deviation or variance in the results would strengthen the argument by demonstrating the consistency of model performance across multiple runs. Additionally, while the paper does a good job supporting its results with evaluation metrics, statistical significance testing between models would further validate the observed differences in performance. Lastly, while the conclusions are well-stated, addressing potential limitations and outlining future directions would provide a more complete discussion of the study's impact and areas for further investigation.

---

## Round 0.2 · accepted · Accept

I thank the Authors for having addressed the Reviewers’ concerns thoroughly.

In particular, (i) the contributions have been clearly presented, (ii) the assessment of the importance of the applied methodology given, (iii) the results and their discussion have been improved, and (iv) statistical tests have been added.

The Reviewers have expressed their satisfaction by accepting the paper, and I have ensured that Reviewer 2’s concerns were addressed thoroughly.

Therefore, the paper is now ready for publication.

Thank you for sending your insightful work to this journal.

Reviewer 1 ·

Basic reporting

fair

Experimental design

fair

Validity of the findings

fair

Additional comments

The authors have successfully addressed all of my comments

Reviewer 3 ·

Basic reporting

The merging of the "Model" and "Method" sections has improved the flow and clarity of the manuscript. The explanation of the RVFL model is now more concise and accessible, which enhances readability. Overall, the manuscript is now well-organized and effectively communicates the research.

Experimental design

The additional details about the computing environment are helpful for reproducibility. I also appreciate the expanded explanation of the data imputation methods and their impact on model performance, which strengthens the methodological section. These revisions make the experimental design more transparent and robust.

Validity of the findings

The inclusion of variance and standard deviation reporting improves the robustness of the results, and the statistical significance tests add rigor to the findings. The enhanced discussion of limitations and future directions also adds context, making the conclusions more comprehensive.